# Human breast tumor-infiltrating CD8+ T cells retain polyfunctionality despite PD-1 expression

Colt A. Egelston [1], Christian Avalos [1], Travis Y. Tu [1], Diana L. Simons[1], Grecia Jimenez[1], Jae Y. Jung[2], Laleh Melstrom[3], Kim Margolin [4], John H. Yim[3], Laura Kruper[3], Joanne Mortimer[4] & Peter P. Lee[1]

Functional CD8+ T cells in human tumors play a clear role in clinical prognosis and response to immunotherapeutic interventions. PD-1 expression in T cells involved in chronic infections and tumors such as melanoma often correlates with a state of T-cell exhaustion. Here we interrogate CD8+ tumor-infiltrating lymphocytes (TILs) from human breast and melanoma tumors to explore their functional state. Despite expression of exhaustion hallmarks, such as PD-1 expression, human breast tumor CD8+ TILs retain robust capacity for production of effector cytokines and degranulation capacity. In contrast, melanoma CD8+ TILs display dramatic reduction of cytokine production and degranulation capacity. We show that CD8+ TILs from human breast tumors can potently kill cancer cells via bi-specific antibodies. Our data demonstrate that CD8+ TILs in human breast tumors retain polyfunctionality, despite PD-1 expression, and suggest that they may be harnessed for effective immunotherapies.

---

[1] Department of Immuno-Oncology, Beckman Research Institute of City of Hope, Duarte, CA 91010, USA. [2] Department of Dermatologic Oncology, Norton Cancer Institute, Louisville, KY 40202, USA. [3] Department of Surgery, Beckman Research Institute of City of Hope, Duarte, CA 91010, USA. [4] Department of Medical Oncology, Beckman Research Institute of City of Hope, Duarte, CA 91010, USA. Correspondence and requests for materials should be addressed to P.P.L. (email: plee@coh.org)

Immune checkpoint blockade immunotherapies have demonstrated efficacy in a number of cancer types, including melanoma, non-small-cell lung cancer, renal cell carcinoma, bladder cancer, and Hodgkin's lymphoma[1–5]. Correlative data from these clinical trials clearly point to the role of CD8+ T-cell infiltration into tumors for therapeutic efficacy[6,7]. CD8+ T cells can exert effector function through their capacity to recognize and kill tumor targets[8,9]. Despite their tumor cytolytic capacity, CD8+ tumor-infiltrating lymphocytes (TILs) may lose their functional potential in the presence of chronic antigen undergoing a state known as T-cell exhaustion[10]. This state is described as a general loss of various effector functions, including cytolytic capacity, proliferative capacity, and production of cytokines interferon-γ (IFNγ), tumor necrosis factor-α (TNFα), and interleukin-2 (IL-2)[11,12].

Early work on T-cell exhaustion primarily involved murine models of chronic infection. Dysfunctional T cells have since been identified in human infectious diseases, such as hepatitis B, hepatitis C, and human immunodeficiency virus[13–15]. Additionally, tumor antigen-specific CD8+ T cells with severely reduced function have been described in melanoma patients[16,17]. In both chronic infections and cancer, exhausted CD8+ T cells have been shown to upregulate the expression of the checkpoint molecule programmed cell death protein 1 (PD-1), which has therefore largely been viewed as a surrogate marker of T-cell exhaustion[18–22]. However, it is important to note that PD-1 was first described as a molecule upregulated upon T-cell activation, and exerts inhibitory activity only upon engagement by programmed death-ligand 1 (PD-L1)[23,24]. Furthermore, PD-1 signaling has been shown to be unnecessary for the induction of T cell exhaustion, and instead it has been shown to be critical for the prevention of T cell terminal proliferation and exhaustion through its role in inhibiting T cell receptor mediated signalling[25].

Exhausted T cells include a heterogeneity of T cells in various functional and phenotypic states. Beyond PD-1 expression, exhausted T cells have been described to upregulate a variety of checkpoint molecules, including LAG-3, CD160, 2B4, TIM-3, and TIGIT[26–30]. T-box transcription factors T-bet and Eomesodermin (Eomes) have been found associated with PD-1 intermediate and PD-1 high subsets respectively, with PD-1hiEomeshi defining greater functional exhaustion[31,32].

Loss of IL-7 receptor-α (CD127), a protein critical for T-cell homeostasis, is observed on T cells with the most extensively exhausted phenotype[19,33–35]. Expression patterns of CD127 together with killer cell lectin-like receptor subfamily G member 1 (KLRG1) can be used to analyze T-cell differentiation states with distinct responses to acute and chronic antigen in the generation of effector cells, contraction of memory cells, and terminally exhausted cells[36]. Short-lived, effector cells express KLRG1 and lack CD127, while conversely long-lived memory cells and their precursors express CD127 and lack KLRG1. While a CD127− KLRG1− phenotype is found on early effector cells after initial antigen exposure, it is also found on severely exhausted terminal effector cells resulting from chronic antigen exposure[37,38].

In breast cancer, presence of TILs is predictive of response to chemotherapy and associates favorably with patient survival[39–41]. Despite this, clinical responses to anti-PD-1 or anti-PD-L1 antibodies in breast cancer patients have been modest with lower objective response rates and shorter response durations compared to those seen in neoplasms such as melanoma[42–46]. A better understanding of CD8+ T-cell composition and functional state would benefit future design of immunotherapeutic trials for breast cancer patients. Here we describe detailed phenotypic and functional profiling of human breast cancer tumor-infiltrating CD8+ T cells. We demonstrate that despite PD-1 expression,

these T cells retain potent functional capacity, including degranulation and production of IFNγ, TNFα, and IL-2. Additionally, these CD8+ TILs retain the ability to kill target cells when redirected with a bi-specific antibody. These results caution against the indiscriminate use of PD-1 as a marker for T-cell exhaustion for all tumor types.

## Results

**bcTumor CD8+ TILs are predominantly effector memory cells.** To study the composition of CD8+ TILs in human breast cancer patients, we obtained primary tumor tissue (bcTumor) and peripheral blood mononuclear cells (bcPBMCs) from a cohort of 61 breast cancer patients undergoing standard surgical procedures (Supplementary Table 1). The cohort was primarily composed of tissues, largely estrogen receptor-positive (ER+) primary tumors. For comparative studies, PBMCs were also obtained from age-matched female healthy donors (hPBMCs). Since melanoma tumors have been well described by others as immunogenic and often containing populations of exhausted T cells, we obtained melanoma metastatic tumor tissue (melTumor) (Supplementary Table 2) for comparison studies to bcTumor tissue.

Analysis of CD8+ T cells by flow cytometry was carried out for memory T-cell subsets. Cells were identified as naive (CCR7+ CD45RA+), central memory (CM, CCR7+ CD45RA−), effector memory (EM, CCR7− CD45RA−), or effector memory RA+ (EMRA, CCR7− CD45RA+) (Fig. 1a). bcTumor and melTumor CD8+ TILs were both primarily composed of EM cells (mean 70%, 86%) (Fig. 1b). The frequency of naive CD8+ T cells was sharply diminished in bcTumor compared to bcPBMCs. melTumor CD8+ TILs were similar in composition to bcTumor CD8+ TILs, but had an even lower frequency of CM cells. In contrast, bcPBMC CD8+ T cells were composed of large populations of naive, EM, and EMRA cells at similar frequencies to hPBMCs (Supplementary Fig. 1). These data clearly point to the preferential accumulation of antigen experienced effector memory cells in tumor tissue.

**bcTumor CD8+ TILs have elevated frequencies of PD-1+ cells.** We characterized CD8+ TILs in these tissues for the presence of various checkpoint molecules which have been shown to critically affect T-cell function in the tumor microenvironment. To do so we analyzed antigen experienced (CD45RA−) CD8+ T cells for expression of PD-1, TIGIT, 2B4, BTLA, TIM-3, LAG-3, and CD160 (Fig. 2). PD-1, TIGIT, and 2B4 were the dominant checkpoint molecules expressed on bcTumor CD45RA− CD8+ TILs (mean 74%, 72%, 66%). Importantly, a large percentage of circulating CD8+ T cells in bcPBMCs also expressed PD-1, TIGIT, and 2B4 at high frequencies (mean 44%, 53%, 52%). Of the various checkpoints examined, only the frequency of PD-1+ cells in bcTumor CD8+ TILs was significantly higher than in bcPBMC CD8+ T cells. We observed no differences in checkpoint molecule expression between CD8+ T cells from bcPBMCs or hPBMCs (Supplementary Fig. 2).

Similar to bcTumor CD8+ TILs, melTumor CD8+ TILs predominantly expressed PD-1, TIGIT, and 2B4 (mean 82%, 82%, 90%). Both 2B4 and TIM-3 were found on a significantly higher frequency of CD8+ TILs in melTumor compared to bcTumor. Thus, although melTumor and bcTumor share similar characteristics of high frequencies of PD-1 and TIGIT expressing CD8+ TILs, expression of additional checkpoint molecules differentiates melTumor CD8+ TILs from bcTumor CD8+ TILs.

**bcTumor CD8+ TILs have elevated frequencies of PD-1+ Eomes+ cells.** T-bet has been shown to repress expression of PD-1 and therefore demonstrates inverse expression to PD-1[47].

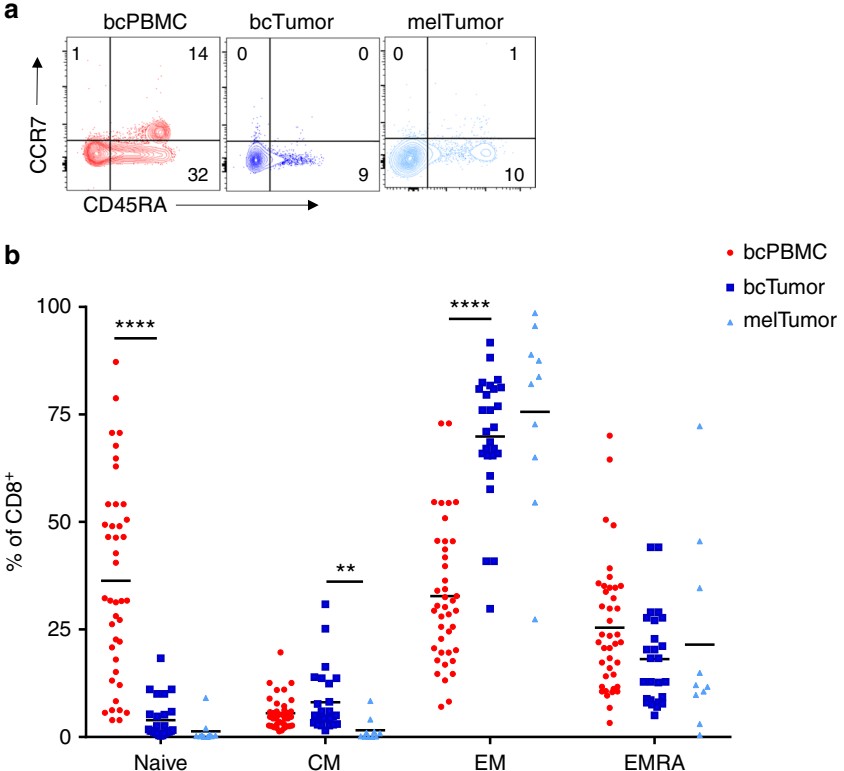

**Fig. 1** CD8[+] TILs in breast tumors are predominantly effector memory cells. **a** Memory phenotypes of CD8[+] T cells in patient peripheral blood mononuclear cells (bcPBMCs), breast cancer patient primary tumors (bcTumor), and melanoma patient tumors (melTumor) were phenotypically characterized by flow cytometry. Representative contour plots show cell population gates for naive (CCR7[+] CD45RA[+]), central memory (CM, CCR7[+] CD45RA[−]), effector memory (EM, CCR7[−] CD45RA[−]), or effector memory RA[+] (EMRA, CCR7[−] CD45RA[+]). **b** Graph depicts percentage of naive, CM, EM, and EMRA among CD8[+] T cells in bcPBMCs, bcTumor and melTumor. Each symbol represents data from a unique patient sample. Significance was calculated using one-way ANOVA and Holm–Sidak multiple comparison tests; **$p < 0.01$; ****$p < 0.0001$

Furthermore, loss of T-bet expression and a PD-1[hi] Eomes[+] phenotype has been shown to correlate with functional exhaustion[31,32]. We therefore next examined CD8[+] T cells for Eomes and T-bet expression patterns and the presence of PD-1[+] Eomes[+] within non-naive CD8[+] T cells from bcPBMCs, bcTumor, and melTumor (Fig. 3a). bcTumor CD8[+] TILs showed a higher frequency of Eomes[+] T-bet[−] cells and a lower frequency Eomes[+] T-bet[+] cells compared to bcPBMC CD8[+] T cells (Fig. 3b). melTumor CD8[+] TILs exhibited a similar expression pattern of Eomes and T-bet as bcTumor CD8[+] TILs, but surprisingly trended towards having a higher frequency of Eomes[+] T-bet[+] cells.

CD8[+] TILs from bcTumor were composed of twice the frequency of PD-1[+] Eomes[+] cells compared to bcPBMC CD8[+] non-naive T cells (Fig. 3c). melTumor CD8[+] TILs also showed a similar, though more varied, fraction of PD-1[+] Eomes[+] cells to bcTumor. We examined levels of PD-1 expression on bcTumor CD8[+] TILs and bcPBMC CD8[+] T cells matched from the same patient (Fig. 3d). PD-1 levels were significantly elevated on bcTumor CD8[+] TILs compared to the bcPBMC CD8[+] T cells in every patient. Together, these data demonstrate that bcTumor CD8[+] TILs display a reduced frequency of T-bet expressing cells as compared to circulating CD8[+] memory T cells and as a result an increased frequency of PD-1[+] Eomes[+] cells.

**Differentiation status of bcTumor CD8[+] TILs.** We hypothesized that in the presence of chronic antigen and inflammation, CD8[+] TILs would be composed of more terminally differentiated T cells relative to circulating CD8[+] T cells. To investigate the

differentiation status of CD45RA[−] CD8[+] T cells in bcTumor, bcPBMC, and melTumor we analyzed the expression of KLRG1 and CD127 in non-naive CD45RA[−] CD8[+] T cells (Fig. 4a). Short-lived effector cells express KLRG1 and lack CD127 and long-lived memory cells express CD127 and lack KLRG1. Loss of CD127 and KLRG1 is described on T cells in conditions of severely terminal exhaustion resulting from exposure to chronic antigen[37,38].

bcTumor CD8[+] TILs had a significantly higher frequency of CD127[−] KLRG1[−] cells compared to a near absence of such cells in bcPBMC CD8[+] T cells (Fig. 4b). Moreover, in melTumor CD8[+] TILs CD127[−] KLRG1[−] cells were even more significantly frequent than in bcTumor and were the predominant population. Although displaying a similar population frequency of CD127[−] KLRG1[+] effector cells as bcPBMC CD8[+] T cells, bcTumor CD8[+] TILs did have a significantly lower frequency of CD127[+] KLRG1[+] cells. This double positive population has previously been shown to denote memory cells re-exposed to antigen and are abundant in the circulation[48].

Since PD-1 expression reflects recent antigen encounter, we hypothesized that PD-1[+] CD8[+] T cells would be composed of more terminally differentiated T cells compared to PD-1[−] CD8[+] T cells. We compared the expression of CD127 and KLRG1 within PD-1[+] and PD-1[−] CD8[+] T cells from bcPBMCs, bcTumor, and melTumor. PD-1[+] CD8[+] TILs from bcTumor and especially melTumor contained higher frequencies of CD127[−] KLRG1[−] cells compared to PD-1[−] CD8[+] TILs (Fig. 4c). bcTumor PD-1[+] CD8[+] TILs maintained comparable population frequencies of CD127[−] KLRG1[+], CD127[+] KLRG1[+], and CD127[+] KLRG1[−] cells to bcTumor PD-1[−] CD8[+] TILs.

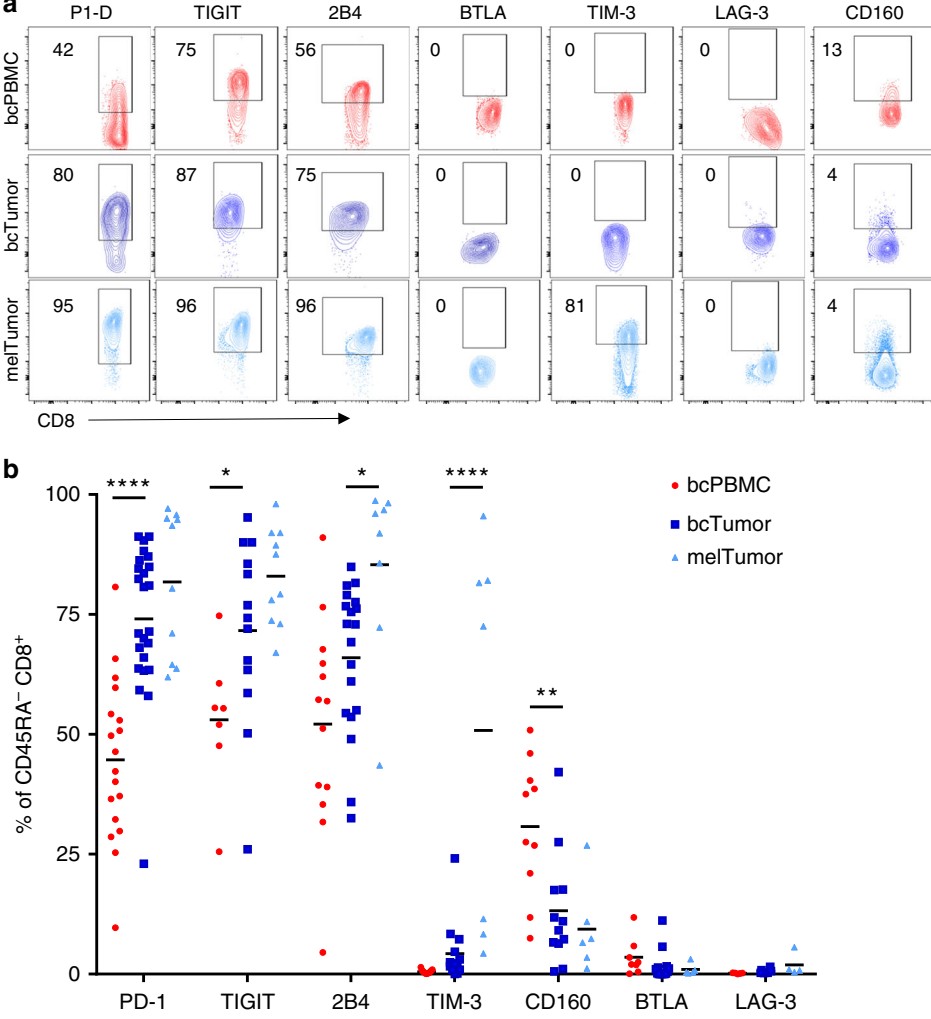

**Fig. 2** CD8+ TILs in breast tumors predominantly express checkpoint molecules PD-1, TIGIT, and 2B4. **a** CD8+ T cells from bcPBMCs, bcTumor, and melTumor were assessed for expression of various checkpoint molecules by flow cytometry as shown in representative contour plots. **b** Graph depicts percentage of non-naive CD45RA− CD8+ T cells that express a given checkpoint molecule. Each symbol represents data from a unique patient sample. Significance was calculated using one-way ANOVA and Holm–Sidak multiple comparison tests; *p < 0.05; **p < 0.01; ****p < 0.0001

However, melTumor PD-1+ CD8+ TILs showed significantly lower frequencies of those same populations compared to melTumor PD-1− CD8+ TILs due to the predominance of a highly homogenous population of PD-1+ CD127− KLRG1− terminally differentiated cells. In contrast, bcTumor CD8+ TILs are more heterogeneously diverse in differentiation expression patterns. Interestingly, PD-1+ CD8+ T cells and PD-1− CD8+ T cells in bcPBMCs were similar in their differentiation state distribution except for a near total absence of CD127− KLRG1− cells in the PD-1+ cells (Supplementary Fig. 3).

**bcTumor PD-1+ CD8+ TILs retain cytokine production capacity**. Identification of bcTumor CD8 TILs with a PD-1+ Eomes+ phenotype led us to the hypothesis that bcTumor CD8+ TILs would be functionally deficient as characteristic of exhausted T cells. We compared antigen experienced T cells from various tissue sources for their IFNγ, TNFα, and IL-2 production capacity (Fig. 5a). Using a polyfunctionality index calculation formula to account for the combinatorial production of IFNγ, TNFα, and IL-2, we numerically evaluated cytokine production by CD8 T+ cells (Fig. 5b)[49]. bcTumor CD8+ TILs demonstrated a significantly greater cytokine production capacity than melTumor CD8+ TILs

and surprisingly bcPBMC CD8+ T cells as well. This reduction in polyfunctional cytokine production capacity in melTumor CD8+ TILs was marked by a loss in TNFα and IL-2 production, while approximately half of bcTumor CD8+ TILs were capable of producing two or more cytokines (Fig. 5c). Importantly, the differences in CD8+ TIL polyfunctionality capacity does not appear to be driven by differences in age or stage of patients analyzed (Supplemental Fig. 5) or by lower ratios of CD8+ T cells to potential suppressor cells such as CD4+ FOXP3+ regulatory T cells or myeloid cells (Supplemental Fig. 6). Since metastatic breast cancer is a significant area of need for therapeutic intervention, we additionally examined several tumor-invaded lymph nodes from breast cancer patients (bcT+LN). CD8+ TILs from these metastatic tissue sites showed a similar retention of polyfunctionality as those from bcTumor, demonstrating that even in breast cancer metastatic tissue sites, CD8+ TILs are largely functional. Interestingly, CD8+ T cells from tumor-draining lymph nodes from melanoma patients and breast cancer patients had similar and non-reduced polyfunctional capacity (Supplemental Fig. 5c). Of note, polyfunctional cytokine capacity of bcPBMC CD8+ T cells was not diminished in comparison to age-matched healthy control PBMC CD8+ T cells (Supplementary Fig. 4).

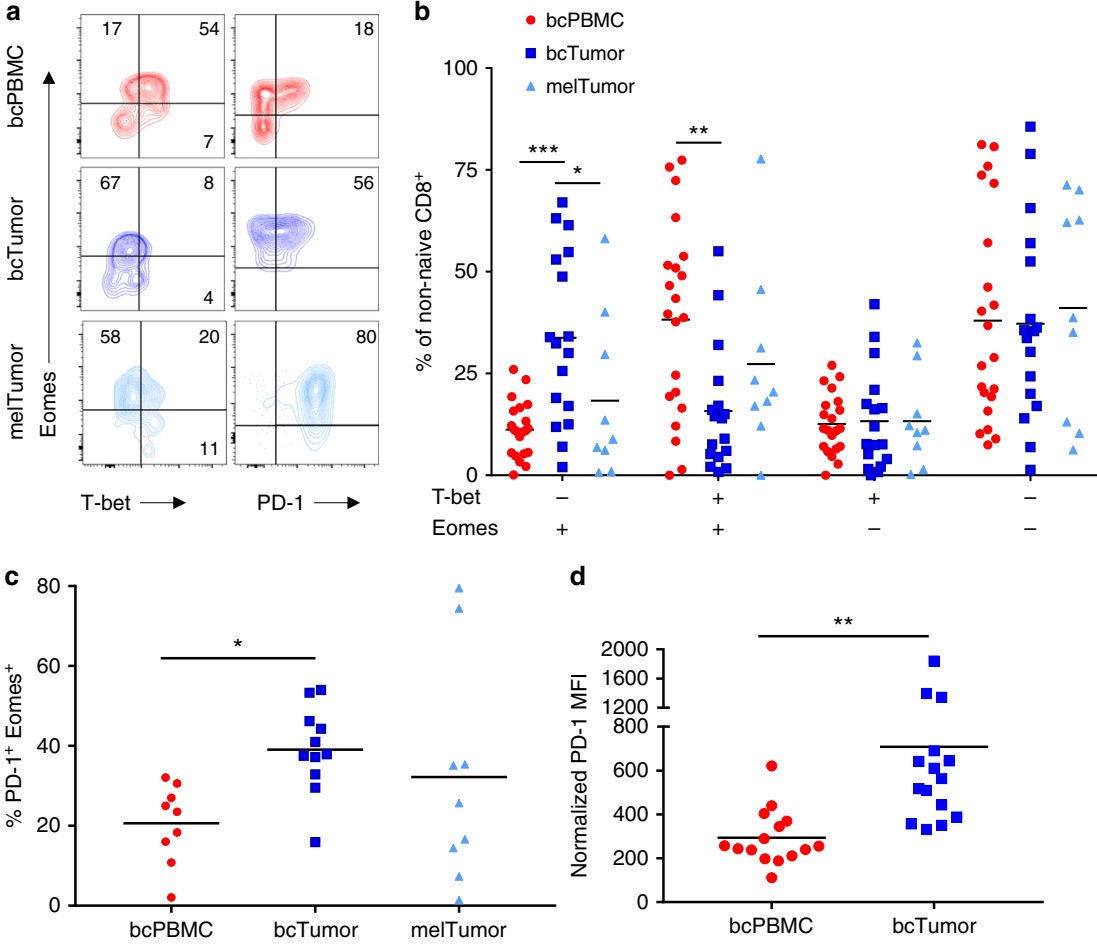

**Fig. 3** CD8$^+$ TILs in breast tumors have elevated frequencies of PD-1$^+$ EOMES$^+$ cells. **a** CD8$^+$ T cells from bcPBMCs, bcTumor, and melTumor were analyzed by flow cytometry for expression of T-bet, Eomes, and PD-1 as shown in representative contour plots. **b** Graph depicts the percentage of Eomes and T-bet expressing populations of non-naive CD8$^+$ T cells in bcPBMCs, bcTumor, and melTumor. **c** Non-naive CD8$^+$ T cells were analyzed for expression of Eomes and T-bet and gated as shown in representative contour plots. **c** Graph depicts the percentage of non-naive CD8$^+$ T cells composed of PD-1$^+$ Eomes$^+$ cells in bcPBMCs, bcTumor, and melTumor. **d** Normalized values of PD-1 expression on non-naive CD8$^+$ T cells in matching bcPBMC and bcTumor samples are shown. Each symbol represents data from a unique patient sample. Significance was calculated using one-way ANOVA and Holm–Sidak multiple comparison tests **b**, **c** or an unpaired Student's $t$-test (**d**); *$p < 0.05$; **$p < 0.01$; ***$p < 0.001$

To determine specifically if PD-1$^+$ T cells in bcTumor maintained polyfunctional capacity we gated on PD-1$^+$ and PD-1$^-$ CD8$^+$ T cells in bcTumor samples and compared cytokine production. PD-1$^+$ CD8$^+$ TILs in bcTumor demonstrated a similar distribution of cytokine expressing subsets to PD-1$^-$ CD8$^+$ TILs (Fig. 6a). Quantification of these subset distributions further demonstrated no decreased polyfunctional capacity to produce effector cytokines in PD-1$^+$ CD8$^+$ TILs (Fig. 6b). Since CD8$^+$ TILs from melTumors displayed a predominantly CD127$^-$ KLRG1$^-$ phenotype, we explored if polyfunctionality of bcTumors CD8$^+$ TILs negatively correlated with the fraction of CD8$^+$ T cells with the same phenotype (Supplementary Fig. 7). No significant association with bcTumor CD8$^+$ T cells polyfunctionality and a CD127$^-$ KLRG1$^-$ phenotype was identified. Furthermore we found that bcTumor CD127$^-$ CD8$^+$ TILs had similar polyfunctional capacity as bcTumor CD127$^+$ CD8$^+$ TILs, but significantly greater capacity than melTumor CD8$^+$ TILs. These data suggest that in addition to PD-1 expression, a CD127$^-$ KLRG1$^-$ phenotype alone does not implicate CD8$^+$ T-cell exhaustion.

**bcTumor CD8$^+$ TILs retain cytotoxic capacity.** Cytolytic capacity is another hallmark of CD8$^+$ T-cell function. To

interrogate the intrinsic ability of bcTumor CD8$^+$ TILs to degranulate and kill target cells we utilized CD3:CD19 bi-specific antibodies to redirect patient isolated CD8$^+$ T cells to attack CD19 expressing target cells without needing T-cell receptor antigen specificity. Using this system we examined the degranulation potential of PD-1$^+$ and PD-1$^-$ CD8$^+$ T cells from bcPBMCs, bcTumor, and melTumor (Fig. 7a). bcTumor PD-1$^+$ CD8$^+$ T cells and PD-1$^-$ CD8$^+$ T cells both degranulated at a similar frequency as measured by a CD107 mobilization assay (Fig. 7b). bcPBMC non-naive PD-1$^+$ CD8$^+$ T cells and PD-1$^-$ CD8$^+$ T cells also degranulated at a similar frequency to each other. Additionally, there was no difference in degranulation between bcPBMCs and bcTumor CD8$^+$ T cells. However, bcTumor PD-1$^+$ CD8$^+$ T cells did show a significantly higher level of degranulation compared to melTumor PD-1$^+$ CD8$^+$ T cells, further demonstrating T-cell exhaustion in melanoma TIL CD8$^+$ T cells and not breast cancer TIL CD8$^+$ T cells.

Finally, we compared the capacity of bcTumor CD8$^+$ TILs and bcPBMC CD45RA$^-$ CD8$^+$ T cells to kill target cells again utilizing the CD3:CD19 bi-specific antibodies. bcTumor CD8$^+$ TILs killed target cells just as efficiently as bcPBMC CD8$^+$ T cells, impressively killing nearly 40% of target cells at a 1:1 effector/ target ratio in an overnight assay (Fig. 7c). This reduction in

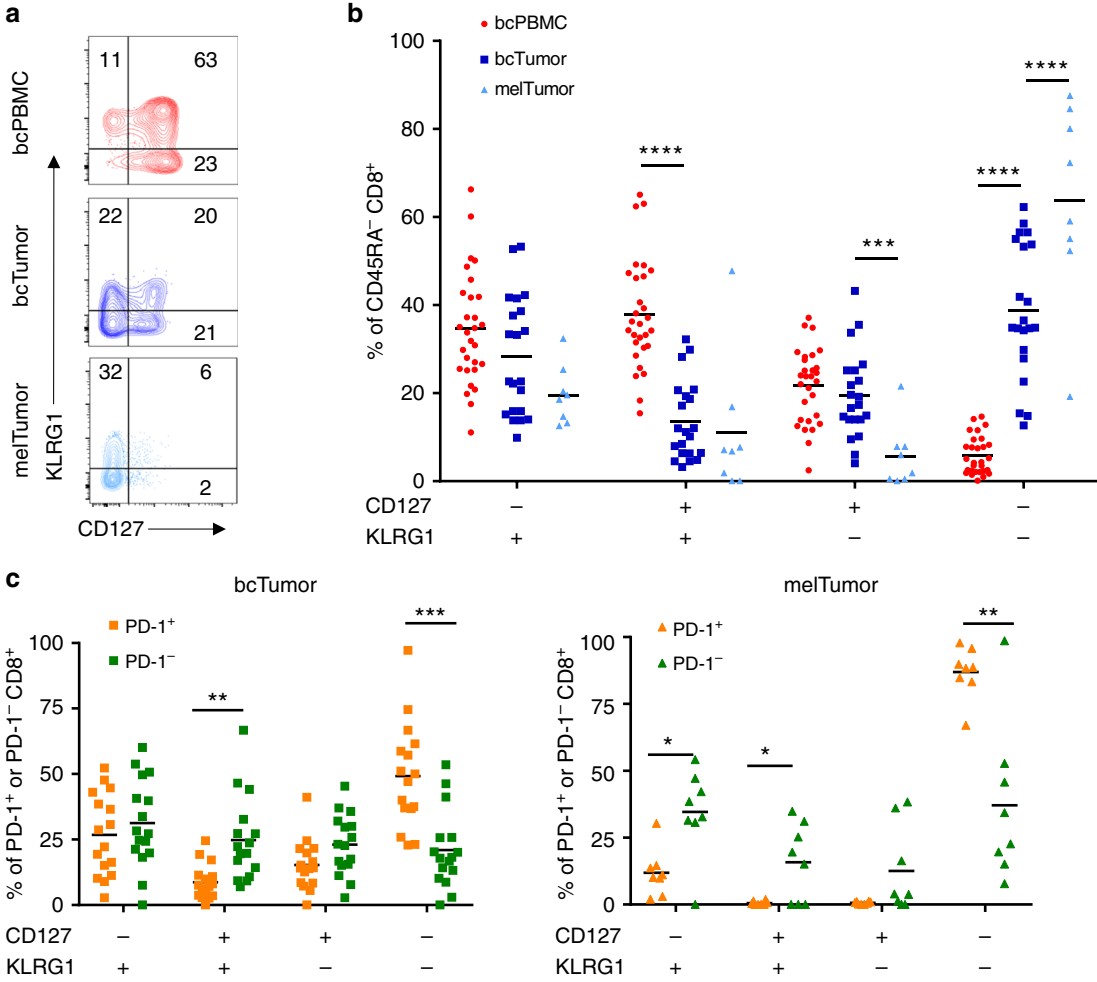

**Fig. 4** Higher frequencies of terminally differentiated CD127⁻ KLRG1⁻ CD8⁺ TILs are present in melanoma tumors than breast tumors. **a** CD8⁺ T cells from bcPBMCs, bcTumor, and melTumor were assessed for expression of CD127 and KLRG1 by flow cytometry as shown in representative contour plots. **b** Graph depicts percentage of non-naive CD45RA⁻ CD8⁺ T cells that express a given expression pattern of CD127 and KLRG1. **c** Graphs depict the proportions of PD-1⁺ and PD-1⁻ CD8⁺ T cells from bcTumor and melTumor with a given expression of CD127 and KLRG1. Each symbol represents data from a unique patient sample. Significance was calculated using one-way ANOVA and Holm–Sidak multiple comparison tests; *$p < 0.05$; **$p < 0.01$; ***$p < 0.001$; ****$p < 0.0001$

target cells was driven by a reduction in absolute target cells alive at the end of co-culture and not by reduced numbers of proliferating target cells, demonstrating a cytotoxic effect by bcTumor CD8⁺ TILs (Supplementary Fig. 8). These data suggest bcTumor CD8⁺ TILs retain cytotoxic potential and bi-specific antibodies may be an effective immunotherapy in breast cancer patients.

## Discussion
In this study, we show that CD8⁺ TILs in human breast tumors are predominantly of effector memory phenotype, express checkpoint molecules PD-1, TIGIT, and 2B4, and are enriched for PD-1⁺ Eomes⁺ T-bet⁻ cells. Despite PD-1 expression, bcTumor CD8⁺ TILs maintain cytokine production capacity and retain the ability to degranulate and kill target cells. Thus, PD-1⁺ CD8⁺ TILs from bcTumor retain polyfunctionality; in contrast, PD-1⁺ CD8⁺ TILs from melTumor are functionally exhausted. These data demonstrate that PD-1 expression should not be utilized as an indiscriminate marker of T-cell exhaustion in all tumor types. Rather, PD-1⁺ CD8⁺ TILs in breast tumors represent potent effector cells that may be harnessed and redirected for tumor cell killing with immunotherapies such as bi-specific antibodies.

The presence of CD8⁺ TILs significantly associates with improved patient prognosis in some but not all of the molecular subtypes of breast cancer. These subtypes are defined by over-expression of ER, progesterone receptor (PR), and receptor tyrosine-protein kinase erbB-2 (HER2)[50]. For reasons that are not yet fully understood, CD8⁺ TILs benefit patients with ER−, ER+ HER2+, and triple-negative tumors, but not ER+ HER2− tumors. As our study was composed of mostly ER+ HER2− breast tumors ($n = 52$), we can conclude the lack of prognostic benefit is not due to a loss of CD8⁺ T-cell functionality in these tumors. While we did not have sufficient sample sizes from each of the molecular subgroups for statistical analysis, we did not recognize differential functional phenotypes between the various molecular subtypes we analyzed. In melanoma patients, the presence of exhausted T cells has been shown to predict clinical response to checkpoint blockade[51]. Our data demonstrating a low frequency of breast tumors with exhausted TILs may explain the lack of success for checkpoint therapies in clinical trials for breast cancer patients. Future studies on tumor antigen specificity and further functional phenotyping of TILs are needed to fully understand the role of TILs in breast cancer patients.

PD-1 is upregulated on T cells upon activation via the T-cell receptor pathway[23,24]. However, in the context of chronic disease

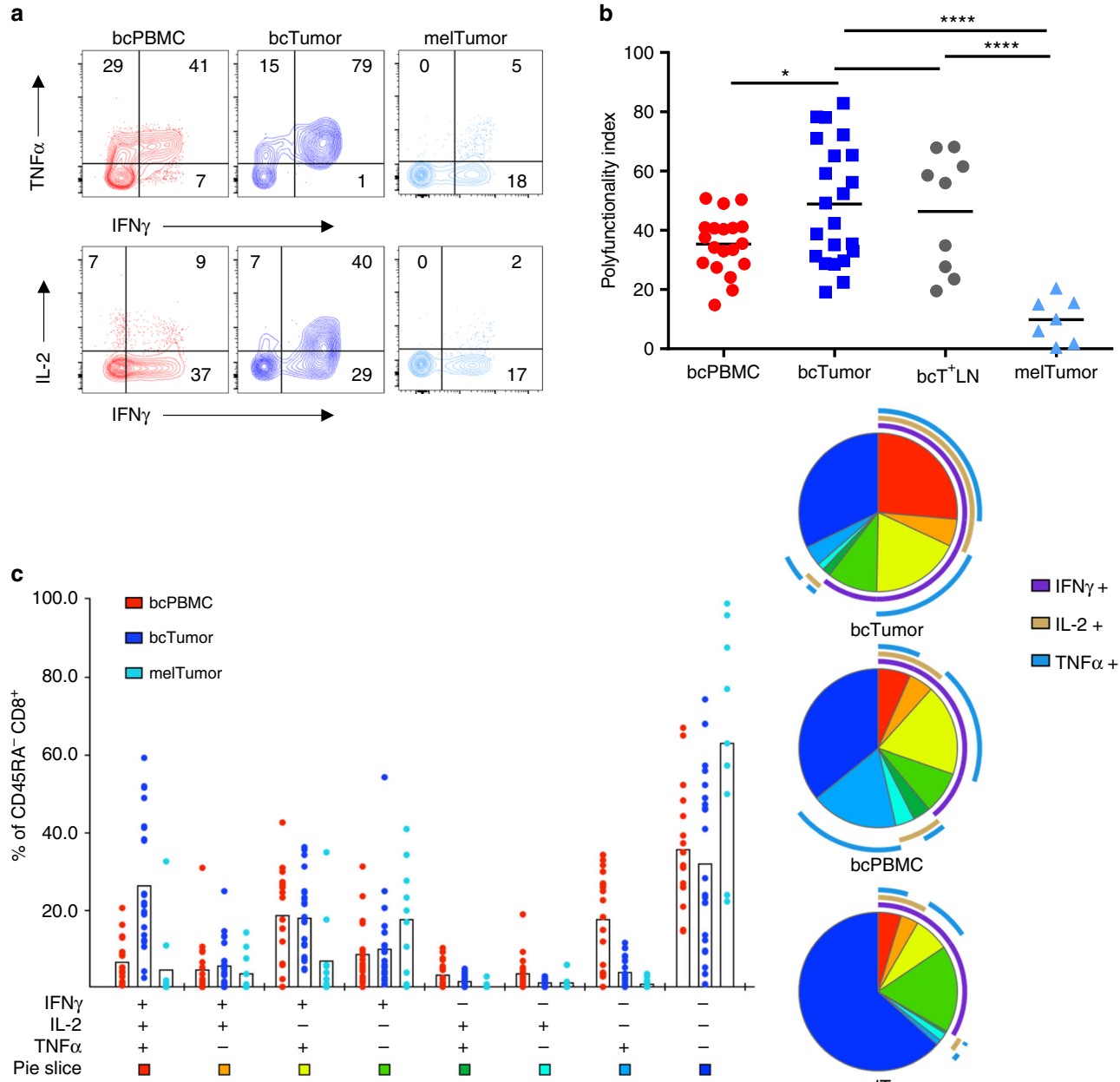

**Fig. 5** CD8+ TILs from breast tumors retain polyfunctional effector cytokine production capacity for cytokines. **a** CD8+ T cells from bcPBMCs, bcTumor, bcT+LN, and melTumor tissues were stimulated with PMA and ionomycin for 4 h followed by intracellular staining for IFNγ, TNFα, and IL-2 production. Representative contour plots of flow cytometry data are shown. **b** Graph depicts calculated polyfunctionality indices for each sample type based on frequencies of non-naive CD8+ T cells expressing IFNγ, TNFα, and IL-2. Each symbol represents a single patient tissue sample. **c** SPICE graph visualizes frequencies of non-naive CD8+ T cells expressing IFNγ, TNFα, and IL-2 in bcPBMCs, bcTumor, melTumor with each symbol representing one patient tissue sample in each category. Summary pie graphs generated represent adjusted means for each population color-coded in the bar graph and display the phenotype of IFNγ, TNFα, and IL-2 production in those populations. Significance was calculated using one-way ANOVA and Holm–Sidak multiple comparison tests; *p < 0.05; ***p < 0.001

such as cancer, PD-1 is more commonly viewed as a marker of T-cell exhaustion rather than as an activation marker[18–22]. Our data demonstrate that although exhausted T cells express PD-1, not all PD-1+ T cells are exhausted, even within the tumor micro-environment. Recent work in the context of simian immunode-ficiency virus and acute myeloid leukemia have also raised questions of the specificity of PD-1 for exhausted T cells in other chronic disease settings[52,53]. One explanation for contradictory findings regarding PD-1 expression and T-cell function is tran-scriptional regulation of PD-1 expression. Demethylation of the

PD-1 promoter early on in a T-cell effector response has been shown to allow PD-1 expression long after antigen encounter and in an antigen-independent manner[54,55]. As a result, PD-1 expression may not necessarily reflect a recent or ongoing response to antigen.

Antigen specificity and tumor antigen specificity of TILs in human breast tumors are largely unknown. Bystander T cells, which lack tumor specificity, have recently been shown as an abundant component of human tumor lymphocyte infiltrate[56]. The retention of polyfunctionality in the majority of CD8+ TILs

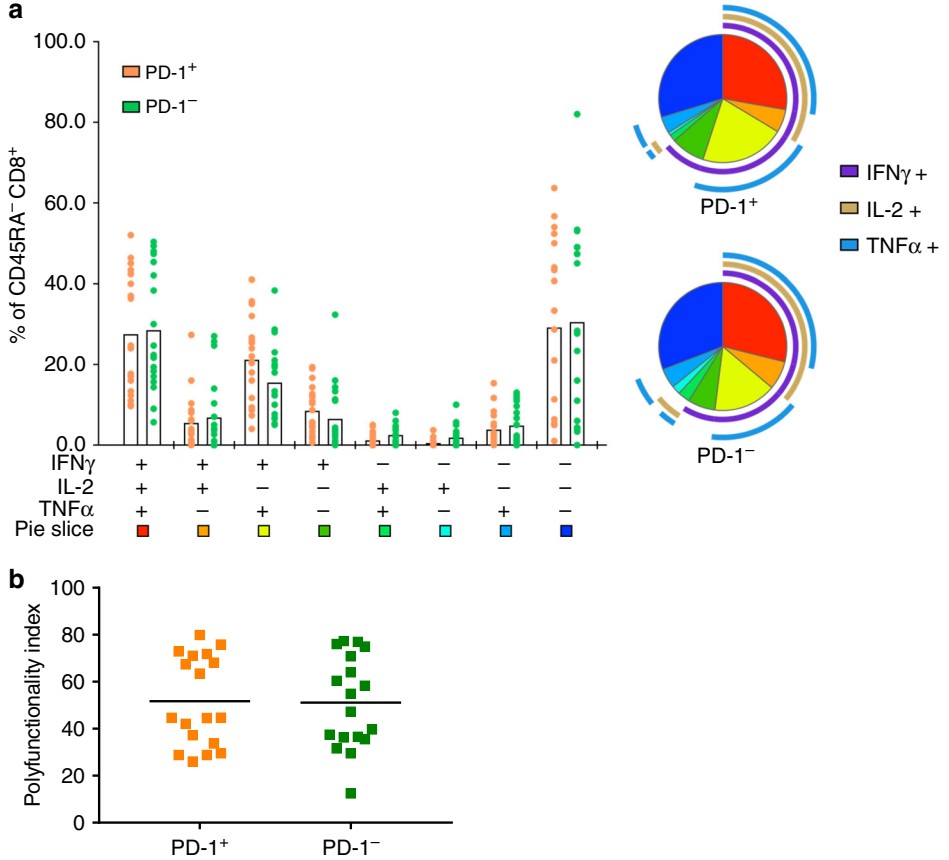

**Fig. 6** PD-1[+] CD8[+] TILs from breast tumors retain polyfunctional effector cytokine production capacity. **a** CD8[+] T cells from bcTumor tissues were stimulated with PMA and ionomycin for 4 h followed by intracellular staining for IFNγ, TNFα, and IL-2 production and surface staining for PD-1. SPICE graph visualizes frequencies of bcTumor non-naive PD-1[+] CD8[+] T cells and non-naive PD-1[−] CD8[+] T cells expressing IFNγ, TNFα, and IL-2 in bcTumor with each symbol representing one patient tissue sample in each category. Summary pie graphs generated represent adjusted means for each population color-coded in the bar graph and display the phenotype of IFNγ, TNFα, and IL-2 production in those populations. **b** Graph depicts calculated polyfunctionality indices for each sample type based on frequencies of bcTumor non-naive PD-1[+] CD8[+] T cells and non-naive PD-1[−] CD8[+] T cells expressing IFNγ, TNFα, and IL-2. **c** CD8[+] T cells from sentinel lymph node tumor negative (SLN T−) or tumor positive (SLN T+) tissues were stimulated with PMA and ionomycin for 4 h followed by intracellular staining for IFNγ and TNFα. Each symbol represents a single patient tissue sample

in human breast tumors therefore suggests the possibility that they are significantly composed of bystander T cells. Further work examining antigen specificity of breast tumor CD8[+] TILs would shed light on a potential diverse combination of T cells specific for neoantigens, self-antigens, and viral antigens. A lack of tumor specificity by CD8[+] TILs in breast tumors may explain a lack of impressive clinical responses to checkpoint blockade therapies[42]. However, the presence of polyfunctional CD8[+] TILs would allow for successful bi-specific antibody-based therapies mediated by cancer cell killing as we preliminarily demonstrate.

Based on these data, a PD-1[hi] Eomes[+] T-bet[−] phenotype does not necessarily implicate a CD8[+] TIL to be in a state of exhaustion. Although both Eomes and T-bet are critical early in T-cell differentiation for proper effector function, Eomes acts to promote memory T-cell formation and T-bet acts to promote terminal effector cell formation[57,58]. Eomes may also play an important role in driving T-cell trafficking to peripheral tissue sites such as tumor tissue[31,59]. Further work to understand the disparate role of Eomes in driving T-cell exhaustion and maintaining quiescent T cells in peripheral tissues is needed.

We found that bcTumor CD8[+] TILs have a lower frequency of CD127[−] KLRG1[−] terminally differentiated cells as compared to the high frequency seen in exhausted melTumor CD8[+] TILs. Interestingly, this phenotype is enriched within PD-1[+] cells in both bcTumor and melTumor, but not bcPBMCs, suggesting that

terminal differentiation may occur in tumor tissue. We propose that T cells chronically driven by tumor antigen eventually lose expression of CD127 and KLRG1 becoming terminally differentiated. Further chronic antigen, as in the case of melTumor, may result in loss of production capacity for effector cytokines IFNγ, TNFα, and IL-2 and becoming terminally exhausted T cells.

Although CD160 and LAG-3 have been reported to be expressed on exhausted T cells, we found low frequencies of TILs expressing either in both bcTumor and melTumor. However, we did find 2B4 and especially TIM-3 to be more significantly expressed on exhausted melTumor CD8[+] T cells than on functional bcTumor CD8[+] T cells. Coexpression of PD-1 and Tim-3 has been described before as specifically identifying CD8[+] TILs with severe dysfunction in a murine tumor model[60]. Likewise, our data support that in human tumors a PD-1[+] TIM-3[+] CD127[−] phenotype may more accurately identify exhausted T cells.

While tumor-infiltrating T cells in numerous cancer types display some combination of elevated checkpoint molecules, there has not been consensus on expression pattern or level of expression that correlates with T-cell exhaustion. To our knowledge, this is the first detailed functional characterization of CD8[+] TILs in human solid tumors decoupling the expression of PD-1 from T-cell exhaustion. This work cautions investigators from broad assumptions about T-cell tumor infiltrates across

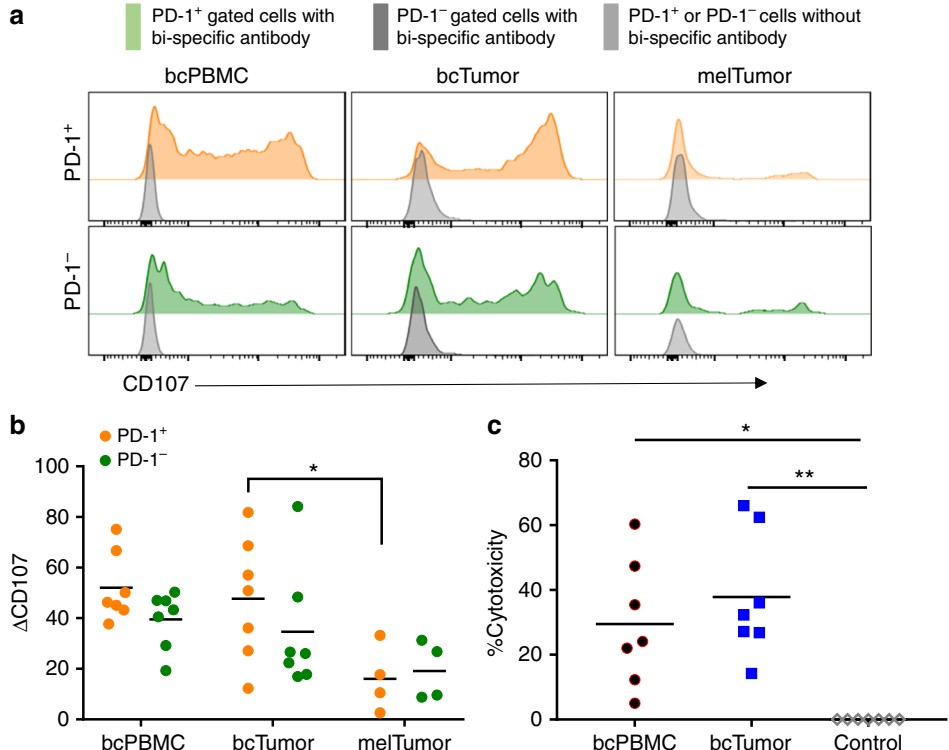

**Fig. 7** PD-1+ CD8+ TILs from breast tumors retain cytotoxic capacity. **a** Non-naive CD8+ T cells isolated from bcPBMCs or bcTumor were cultured with CD19 expressing target cells and CD3-CD19 bi-specific antibodies and analyzed for CD107 mobilization by flow cytometry as shown in representative plots. **b** Graph depicts change (Δ) in frequency of CD107 mobilizing CD8+ cells in co-cultures with bi-specific antibody relative to those without. Parent gates were of PD-1+ and PD-1− populations from bcPBMC and bcTumor CD8+ CD45RA− T cells. **c** Graph depicts cytotoxicity of bcPBMC and bcTumor CD8+ after overnight co-culture with bi-specific antibodies and target cells. Significance was calculated using one-way ANOVA and Holm–Sidak multiple comparison tests; *$p < 0.05$; **$p < 0.01$

cancer types and highlights the need for better functional understanding of TILs in order to develop precision-based immunotherapies for various cancer types. Furthermore, our data highlight the functional potential of bcTumor CD8+ TILs and the potential for bi-specific antibody therapy for human breast cancer patients.

## Methods

**Human samples**. For the purpose of this study we utilized tissues from breast cancer patients ($n = 61$), melanoma patients ($n = 8$), and female age-matched healthy donors ($n = 14$; average age 51 years). Clinical characteristics are summarized in Supplementary Tables 1 and 2. Classification of tumor samples as ER+, PR+, or HER2+ was performed by clinical pathologists. Due to limited cell numbers obtained from patient tumor samples, not all analyses shown were performed on all samples.

**Sample processing**. Patient peripheral blood was obtained by venipuncture using heparin collection tubes, transported at room temperature from the clinic to the lab, and processed within 6 h of drawing. PBMCs were isolated via Ficoll-Paque Separation (GE Healthcare) following the manufacturer's instructions.

Tumor specimens were collected by surgical resection and collected in tubes containing cold RPMI (Life Technologies, Thermo Fisher Scientific) and transported on ice to the laboratory for processing within 1 h of surgery. Tumor tissues were minced into pieces and further mechanically dissociated with a gentleMACS Dissociator (Miltenyi Biotec). Tissue was treated with 0.2 Wunsch U/ml Liberase TM (Roche) and 10 units/ml DNase (Sigma) in RPMI for up to 1 h as needed. If necessary, red blood cell (RBC) lysis was performed using RBC Lysis Buffer (Biolegend).

**Flow cytometry**. Single-cell suspensions were stained at room temperature in 2% fetal bovine serum in phosphate-buffered saline. For cytokine production assays, cells were stimulated with 50 ng/ml phorbol myristate acetate (Sigma) and 1 μg/ml ionomycin (Sigma) in the presence of Golgi Plug (Biolegend) for 4 h. Overnight fixation as needed was performed with IC Fixation Buffer (eBioscience). Fixation and permeabilization was performed with BD Transcription Factor Buffer (BD

Biosciences) or BD Cytofix/Cytoperm buffers as necessary. Antibody cocktails were diluted in Brilliant Violet Buffer (BD Biosciences) when necessary. Samples were acquired using a BD Fortessa using FACS Diva 6.1.3. Photomultiplier tube voltages were set using BD CS&T beads. Compensation was calculated using single-stained OneComp compensation beads (eBioscience). Samples were stained with fluorescently tagged antibodies detailed in Supplementary Table 1. Antibodies were titrated for optimal stain to noise ratio prior to use. Checkpoint molecule-specific antibodies were validated prior to use on in vitro stimulated T cells.

Flow cytometry analysis was performed using Flowjo vX. All samples were gated on single cells, lymphocytes, and CD3+ CD8+ populations. Gates for checkpoint molecules, T-bet, and Eomes were set by gating on naive CD45RA− T cells run in the same experiment. Normalized PD-1 expression was calculated by subtracting median fluorescent intensity of PD-1 staining on PD-1− CD8+ T cells from the PD-1+ CD8+ T cells. Contour plots shown display 5% probability.

**Polyfunctionality index calculation**. The polyfunctionality index equation was applied as described by Larsen et al.[49]. The polyfunctionality index was implemented in R (version 3.3.2) to take SPICE formatted csv files as inputs and output a txt file with the polyfunctionality index of each sample.

**Degranulation and cytotoxicity assays**. CD8+ T cells were isolated with EasySep Human CD8 Positive Selection Kit (Stem Cell Technologies) from bcTumor or EasySep Human Memory CD8+ Enrichment Kit from bcPBMCs. T cells were pre-stained with fluorescent anti-PD-1 antibody immediately before culture setup. The CD19+ B cell line C1R:A2 (kindly donated by the laboratory of Jill Slansky) was used at a 1:1 effector/target ratio. In order to target CD8+ T cells to this cell line, we added bi-specific CD19-directed CD3 T-cell engager Blincyto (Blinatumomab, Amgen, Thousand Oaks, CA) to cultures at a final concentration of 125 ng/ml. Targets and effectors were cultured at 37 °C for 4 h to allow for cytotoxic T lymphocyte (CTL) effector function. Detection of CD107 mobilization was done as previously described[61]. Anti-CD107 antibodies were included at the start of the CTL assay with 7.2 μM of monensin. Change (Δ) in frequency of CD107 mobilizing T cells from co-cultures with and without bi-specific antibodies are reported. For cytotoxicity assays, similar co-culture conditions were utilized but carried out overnight instead. The following day, CD19+ target cells were quantified by flow cytometry using Precision Count Beads (Biolegend). Percent cytotoxicity was

calculated as (no. of control well target cells − no. of target cells in co-culture)/(no. of control well target cells).

**Statistics**. Analysis and presentation of distributions was performed using SPICE version 5.1, downloaded from http://exon.niaid.nih.gov[62]. Graphs and statistics were performed using Graphpad Prism 7.02. Statistics described were generated using unpaired Student's t-tests or one-way analysis of variance (ANOVA) with Holm–Sidak multiple comparison tests. In all relevant graphs, bcPBMCs and melTumor were statistically compared to bcTumor, but not each other. Calculated p values are displayed as $*p < 0.05$; $**p < 0.01$; $***p < 0.001$; $****p < 0.0001$. For all graphs, the mean is represented by a line.

**Study approval**. Fresh tumor and peripheral blood were obtained from patients who gave institutional review board (IRB)-approved written informed consent prior to inclusion in the study (City of Hope IRB 05091, IRB 07047, and IRB 15174).

## Data availability

Data supporting the findings of this study are available within the article and its supplementary information files and from the corresponding author upon reasonable request.

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

# ARTICLE

49. Larsen, M. et al. Evaluating cellular polyfunctionality with a novel polyfunctionality index. *PLoS One* **7**, e42403 (2012).
50. Dai, X. et al. Breast cancer intrinsic subtype classification, clinical use and future trends. *Am. J. Cancer Res.* **5**, 2929–2943 (2015).
51. Loo, K. et al. Partially exhausted tumor-infiltrating lymphocytes predict response to combination immunotherapy. *JCI Insight* **2**, pii: 93433 (2017).
52. Hong, J. J., Amancha, P. K., Rogers, K., Ansari, A. A. & Villinger, F. Re-evaluation of PD-1 expression by T cells as a marker for immune exhaustion during SIV infection. *PLoS One* **8**, e60186 (2013).
53. Schnorfeil, F. M. et al. T cells are functionally not impaired in AML: increased PD-1 expression is only seen at time of relapse and correlates with a shift towards the memory T cell compartment. *J. Hematol. Oncol.* **8**, 93 (2015).
54. Ahn, E. et al. Demethylation of the PD-1 promoter is imprinted during the effector phase of CD8 T cell exhaustion. *J. Virol.* **90**, 8934–8946 (2016).
55. Shwetank, et al. Maintenance of PD-1 on brain-resident memory CD8 T cells is antigen independent. *Immunol. Cell Biol.* **95**, 953–959 (2017).
56. Simoni, Y. et al. Bystander CD8(+) T cells are abundant and phenotypically distinct in human tumour infiltrates. *Nature* **557**, 575–579 (2018).
57. Banerjee, A. et al. Cutting edge: The transcription factor eomesodermin enables CD8+T cells to compete for the memory cell niche. *J. Immunol.* **185**, 4988–4992 (2010).
58. Joshi, N. S. et al. Inflammation directs memory precursor and short-lived effector CD8(+) T cell fates via the graded expression of T-bet transcription factor. *Immunity* **27**, 281–295 (2007).
59. Mackay, L. K. et al. T-box transcription factors combine with the cytokines TGF-beta and IL-15 to control tissue-resident memory T cell fate. *Immunity* **43**, 1101–1111 (2015).
60. Singer, M. et al. A distinct gene module for dysfunction uncoupled from activation in tumor-infiltrating T cells. *Cell* **166**, 1500–1511 e1509 (2016).
61. Rubio, V. et al. Ex vivo identification, isolation and analysis of tumor-cytolytic T cells. *Nat. Med.* **9**, 1377–1382 (2003).
62. Roederer, M., Nozzi, J. L. & Nason, M. C. SPICE: exploration and analysis of post-cytometric complex multivariate datasets. *Cytometry A* **79**, 167–174 (2011).

## Acknowledgements

The authors would like to thank Michele Kirschenbaum for obtaining patient consent and procuring tissue samples. We also thank George Somlo and James Waisman for clinical support. We thank Manasi Kamat for valuable statistics consultation and Silke Lindner, Sabina Muend, and David Rose for critical reading of this manuscript. We especially thank our patient tissue donors and our breast cancer patient advocate Susie Brain. This work was support by the DoD BCRP award W81XWH-11–1–0548. Research reported in this publication included work performed in the Analytical Cytometry Core supported by the National Cancer Institute of the National Institutes of Health under award number P30CA33572. The content is solely the responsibility of the authors and does not necessarily represent the official views of the National Institutes of Health.

## Author contributions

C.A.E., C.A., and P.P.L. designed research studies and conceptualized experiments. C.A.E., C.A., T.Y.T., D.L.S., and G.J. conducted experiments, acquired data, and analyzed data. J.Y.J., L.M., K.M., J.H.Y., L.K., and J.M. provided clinical sample support. C.A.E. and P.P.L. wrote the manuscript.

## Additional information

**Competing interests:** The authors declare no competing interests.

