## [Peer Review File · Nature Communications]

Reviewers' comments:

Reviewer #1:

(Remarks to the Author):

Egelston and colleagues present a manuscript of a study that has evaluated the functionality of CD8 T cells from both blood and tumor of patients with breast cancer. The major finding of the manuscript is that tumor-infiltrating CD8 T cells retain strong function despite demonstrating an 'exhausted' phenotype described as IL-7R-negative, PD-1-positive and Eomes-positive CD8 T cells. The data apparently contrasts with the function of similarly defined CD8 T cells localized in the melanoma microenvironment. Overall, the study is fairly compelling and will make a novel and needed addition to the scientific literature and knowledgebase. Additionally, the paper is nicely crafted and reads well. There are some issues however that are problematic and need to be resolved or discussed. These issues are as follows:

1. The number of melanoma samples is very low and the variability in the outcomes of the various functional and phenotypic assays is wide. A recommendation is that the investigators analyze additional melanoma samples to more solidly support the conclusions.
2. The ages of the cohorts appear to be different and functionality/phenotype could be related to age. The authors should provide some information and/or discussion about how age influences the results of the in vitro studies.
3. Although the authors conclude that the breast tumor-infiltrating CD8 T cells retain cytotoxic activity, the growth assay may not have been the appropriate assay since it might have also detected cytostatic activity too. This concern is offset somewhat by the addition of the CD107 data, but it is not clear if the formal conclusion that there is retention of cytotoxic activity.
4. It is not clear that the CD8 T cells are tumor-antigen-specific T cells. The methods used activate all cells regardless of antigen-specificity. The authors may want to include some data to improve or at very least discuss whether they actually may be bystanders that migrate non-specifically.
5. The conclusion that the melanoma CD8 T cells are functionally exhausted may be disagreeable to some mainly because the functional assays were very limited. It could be argued, due to the use of complex mixtures of cells, rather than purified CD8 T cells, that the melanoma samples had a larger number of regulatory T or myeloid cells that were sufficient to suppress function in the test tube environment. Alternatively, the CD8 T cells may have other functions not assessed by the employed assays, including production of IL-10 or Th2 cytokines, both of which have been reported in the past. Additionally, the investigators did not show evidence of loss of cytotoxic activity in the melanoma infiltrating T cells.

Reviewer #2:

(Remarks to the Author):

In this manuscript, studying samples collected from a large cohort of breast cancer patients, the authors report an important observation that breast TILs retain functions despite PD-1 expression. This might explain partially why the PD-1 targeted agents are not efficient for breast cancer treatment. The article is well written, concept is novel, data are well organized and presented, clinical significance is high. Some limitations are as below:

1. TILs from breast cancer and melanoma samples were compared and the authors concluded that melanoma TILs are exhausted while breast TILs are not. The caveat is that the cancer stage was different between these two groups of patients (supplemental table 1 and 2)-only 1/61 stage IV in

breast but 2/8 in melanoma. It is possible that the late stage of disease status (instead of tumor type) is the factor causing the difference of TILs status (exhaustion vs. not).

2. As expected, variations for phenotypes of TILs are large. It would be interesting to compare the functional difference within the breast cancer patients (e.g. patients with high Tbet-Eomes+% vs. those with low, same in the CD127-KLRG1-). These might be the markers to identify exhausted vs. functional TILs.

3. Correlation with clinical outcome would help to address the clinical application of this information. Eg. Patients with low functional TILs vs. patients with high functional TILs, difference in survival rate? Response to PD-1 treatment (if available)?

4. Number of each studies (n=?) need to be added to the figure legends.

We wish to thank the reviewers for their insightful and helpful comments. We have addressed all issues raised by the reviewers and incorporated changes into the revised manuscript (highlighted in yellow). Following is a detailed description of the modifications that have been made, including a point by point response (in red) to the reviewers' comments.

Reviewer 1:

1. The number of melanoma samples is very low and the variability in the outcomes of the various functional and phenotypic assays is wide. A recommendation is that the investigators analyze additional melanoma samples to more solidly support the conclusions.

Two additional melanoma samples have been analyzed making a total of ten melanoma samples. We would like to stress that the focus of this manuscript is on breast cancer, with melanoma serving as a comparison. Recent published reports have demonstrated that TILs in melanoma are exhausted – consistent with our melanoma data presented here. The primary aim of this manuscript is to show that breast cancer is different by demonstrating polyfunctionality of tumor infiltrating CD8+ T cells in human breast tumors. Additionally per Reviewer 2 comments below, analysis of these melanoma samples is now segregated according to stage as shown in Supplemental Figure 5A, B, and C.

2. The ages of the cohorts appear to be different and functionality/phenotype could be related to age. The authors should provide some information and/or discussion about how age influences the results of the in vitro studies.

We agree that patient age may affect potential polyfunctionality of CD8+ T cells. To assess this, we graphed the polyfunctionality index of tumor infiltrating CD8+ T cells from breast cancer and melanoma patient tumor samples according to groups with ages below or above 55 as shown in Supplemental Figure 5D. Melanoma tumor infiltrating CD8+ T cell polyfunctionality from patients below 55 were significantly lower than those from breast cancer patients below 55. The same was true for samples from patients above the age of 55. Moreover, within breast cancer patients we see no statistical difference in CD8+ T cell polyfunctionality from patients above or below the age of 55.

3. Although the authors conclude that the breast tumor-infiltrating CD8 T cells retain cytotoxic activity, the growth assay may not have been the appropriate assay since it might have also detected cytostatic activity too. This concern is offset somewhat by the addition of the CD107 data, but it is not clear if the formal conclusion that there is retention of cytotoxic activity.

To more formally conclude retention of cytotoxicity by breast cancer CD8+ TILs, we have now included in the manuscript CD107 data from four melanoma tumor samples. We demonstrate that breast tumor CD8+ TILs have a higher capacity for degranulation than melanoma CD8+ TILs. Furthermore, to demonstrate that a reduction of CD19+ target cells in the co-culture assays is a cytotoxic effect and not a cytostatic effect, we have included a graph of the absolute cell numbers at the beginning and end of the co-culture experiment (Supplemental Figure 8). As shown in this graph, co-culture with CD8+ T cells from both breast cancer patient PBMC and TILs led to a reduction in number of surviving target cells from the number of seeded target cells, rather than a reduction in proliferation of target cells.

4. It is not clear that the CD8 T cells are tumor-antigen-specific T cells. The methods used activate all cells regardless of antigen-specificity. The authors may want to include some data to improve or at very least discuss whether they actually may be bystanders that migrate non-specifically.

We agree that antigen-specificity of breast tumor infiltrating CD8+ T cells is an important question and subject of separate investigations. However, the focus of this manuscript is our novel and unexpected finding that infiltrating CD8+ T cells in human breast tumors retain polyfunctionality despite PD-1 expression, in distinct contrast to melanoma – regardless of their antigen specificity. To address this important point, we have added the following paragraph to the discussion section of the manuscript:

“Antigen specificity of TILs in human breast tumors remains largely unknown. Bystander T cells, which lack tumor specificity, have recently been shown as an abundant component of human tumor lymphocyte infiltrate⁵⁶. Retention of polyfunctionality in the majority of CD8⁺ TILs in human breast tumors raises the possibility that they are significantly composed of bystander T cells. Further work examining antigen specificity of breast tumor CD8⁺ TILs would shed light on a potential diverse combination of T cells specific for neoantigens, self-antigens, and viral antigens.”

5. The conclusion that the melanoma CD8 T cells are functionally exhausted may be disagreeable to some mainly because the functional assays were very limited. It could be argued, due to the use of complex mixtures of cells, rather than purified CD8 T cells, that the melanoma samples had a larger number of regulatory T or myeloid cells that were sufficient to suppression function in the test tube environment. Alternatively, the CD8 T cells may have other functions not assessed by the employed assays, including production of IL-10 or Th2 cytokines, both of which have been reported in the past. Additionally, the investigators did not show evidence of loss of cytotoxic activity in the melanoma infiltrating T cells.

To determine the possible effects of regulatory T cells or myeloid cells on CD8⁺ T cell function, we have graphed the CD8/CD4, CD8/Foxp3+CD4, CD8/CD33+ ratios for several breast and melanoma tissues used for this study (Supplementary Figure 6). As the reviewer suggested, higher ratios of CD8⁺ cells to these other cell types could explain the higher functionality seen in breast tumor CD8⁺ TILs. However, we demonstrate that these ratios show no statistical difference for CD8⁺/Treg or CD8⁺/CD33+ ratios. In fact, CD8⁺/CD4 ratios are actually significantly higher in melanoma tumors as compared to breast tumors. We believe this rules out the possibility that the observed differences between CD8⁺ TIL polyfunctionality in breast and melanoma tumors is due to differential suppression exerted by regulatory T cells or myeloid cells.

Reviewer 2:

1. TILs from breast cancer and melanoma samples were compared and the authors concluded that melanoma TILs are exhausted while breast TILs are not. The caveat is that the cancer stage was different between these two groups of patients (supplemental table 1 and 2)-only 1/61 stage IV in breast but 2/8 in melanoma. It is possible that the late stage of disease status (instead of tumor type) is the factor causing the difference of TILs status (exhaustion vs. not).

As described in response to Reviewer 1 comment 1, additional melanoma samples are now included in our analysis and one melanoma sample has been corrected to Stage III that was inaccurately classified as Stage IV in our manuscript previously. Polyfunctional capacity of CD8⁺ TILs has been re-graphed according to breast cancer stage and melanoma stage respectively in a new supplemental figure (Supplementary Figure 5). We show that reduced CD8⁺ T cell polyfunctionality is significantly lower in melanoma tissues than breast tumor tissues regardless of Stage II or Stage III status. Additionally, melanoma lymph node metastases were separated from non-LN metastases for graphing to further highlight the diminished polyfunctionality seen in CD8⁺ TILs from non-LN melanoma metastases as compared to breast tumor CD8⁺ TILs.

2. As expected, variations for phenotypes of TILs are large. It would be interesting to compare the functional difference within the breast cancer patients (e.g. patients with high Tbet-Eomes+% vs. those with low, same in the CD127-KLRG1-). These might be the markers to identify exhausted vs. functional TILs.

We agree that a CD127- KLRG1- phenotype strongly correlated with CD8⁺ T cell exhaustion in melanoma tumors. To assess this in breast cancer tumors we have added the following text and further analysis in Supplementary Figure 7.

“Since CD8⁺ TILs from melTumors displayed a predominantly CD127⁻ KLRG1⁻ phenotype, we explored if polyfunctionality of bcTumors CD8⁺ TILs negatively correlated with the fraction of CD8⁺ T cells with the same phenotype (Supplementary Figure 7). No significant association with bcTumor CD8⁺ T cells polyfunctionality and a CD127⁻ KLRG1⁻ phenotype was identified. Furthermore we found that bcTumor CD127⁻ CD8⁺ TILs had similar polyfunctional capacity as bcTumor CD127⁺ CD8⁺ TILs, but significantly greater capacity than melTumor CD8⁺ TILs. These data suggest that in addition to PD-1 expression, a CD127⁻ KLRG1⁻ phenotype alone does not implicate CD8⁺ T cell exhaustion.”

3. Correlation with clinical outcome would help to address the clinical application of this information. Eg. Patients with low functional TILs vs. patients with high functional TILs, difference in survival rate? Response to PD-1 treatment (if available)?

We suggest that one exciting clinical application of this information is in designing therapies to harness the polyfunctional capacity of CD8⁺ TILs that are already present within many human breast tumors. One possible therapeutic approach is redirected cancer cell killing via bispecific antibodies, as we demonstrated ex vivo in Figure 7C and Supplemental Figure 8. Since most human breast tumors contain polyfunctional TILs, their presence may not predict survival. To address these issues, we have added the following text to the manuscript discussion:

“Lack of tumor specificity by CD8⁺ TILs in human breast tumors may explain the suboptimal clinical responses to checkpoint blockade in breast cancer patients⁴². Instead, the presence of polyfunctional CD8⁺ TILs in human breast tumors could be leveraged for successful bi-specific antibody based therapies mediated by redirected cancer cell killing as we demonstrate ex vivo.”

4. Number of each studies (n=?) need to be added to the figure legends.

The number of samples assayed for each figure has now be included for all data.

REVIEWERS' COMMENTS:

Reviewer#1 (Remarks to the Author):

Editorial note: Reviewer#1 expressed his satisfaction with the revisions in confidential comments to the editor.

Reviewer #2 (Remarks to the Author):

All questions were appropriated addressed.